# Albuminuria in People Chronically Exposed to Low-Dose Cadmium Is Linked to Rising Blood Pressure Levels

**DOI:** 10.3390/toxics13020081

**Published:** 2025-01-23

**Authors:** Soisungwan Satarug, Supabhorn Yimthiang, Tanaporn Khamphaya, Phisit Pouyfung, David A. Vesey, Aleksandra Buha Đorđević

**Affiliations:** 1Centre for Kidney Disease Research, Translational Research Institute, Woolloongabba, Brisbane, QLD 4102, Australia; david.vesey@health.qld.gov.au; 2Occupational Health and Safety, School of Public Health, Walailak University, Nakhon Si Thammarat 80160, Thailand; ksupapor@mail.wu.ac.th (S.Y.); tanaporn.kh@mail.wu.ac.th (T.K.); 3Department of Community Health, Faculty of Public Health, Mahidol University, Bangkok 20100, Thailand; phisit.pou@mahidol.edu; 4Department of Kidney and Transplant Services, Princess Alexandra Hospital, Brisbane, QLD 4102, Australia; 5Department of Toxicology “Akademik Danilo Soldatović”, University of Belgrade-Faculty of Pharmacy, 11000 Belgrade, Serbia; aleksandra.buha@pharmacy.bg.ac.rs

**Keywords:** albuminuria, blood pressure, β_2_-microglobulin, cadmium, estimated glomerular filtration rate, mediation analysis

## Abstract

Exposure to low-dose environmental pollutant cadmium (Cd) increases the risks of both albuminuria and hypertension by mechanisms which are poorly understood. Here, multiple regression and mediation analyses were applied to data from 641 Thai subjects of whom 39.8%, 16.5%, 10.8%, and 4.8% had hypertension, albuminuria, diabetes, and chronic kidney disease (CKD), defined as the estimated glomerular filtration rate (eGFR) ≤ 60 mL/min/1.73 m^2^, respectively. To correct for interindividual differences in urine dilution and surviving nephrons, the excretion rates of Cd (E_Cd_), albumin (E_alb_), and β_2_-microglobulin (E_β2M_) were normalized to the creatinine clearance (C_cr_) as E_Cd_/C_cr_, E_alb_/C_cr_, and E_β2M_/C_cr_. The respective risks of having CKD and hypertension rose to 3.52 (95% CI: 1.75, 7.05) and 1.22 (95% CI: 1.12, 1.3) per doubling of the Cd body burden. The respective risk of having albuminuria increased 2.95-fold (*p* = 0.042) and 4.17-fold (*p* = 0.020) in subjects who had hypertension plus severe and extremely severe tubular dysfunction, defined according to the elevated β_2_M excretion rates. In multiple regression analysis, the E_alb_/C_cr_ increased linearly with both the systolic blood pressure (SBP, β = 0.263) and diastolic blood pressure (DBP, β = 0.150), while showing an inverse association with eGFR (β = −0.180). The mediation model analyses inferred that a declining eGFR induced by Cd contributed to 80.6% of the SBP increment (*p* = 0.005), which then fully mediated an elevation of albumin excretion (*p* < 0.001). The present study provides, for the first time, evidence that causally links Cd-induced eGFR reductions to blood pressure elevations, which enhance albumin excretion.

## 1. Introduction

Approximately 8–13% of the world’s population lives with chronic kidney disease (CKD) [1,2,3]. CKD is predicted to become the fifth leading cause of years of life lost by 2040 [4,5]. In the early stages, CKD is asymptomatic, and it is diagnosed when there is a substantial loss of functioning nephrons, evident from a fall in the estimated glomerular filtration rate (eGFR) below 60 mL/min/1.73 m^2^ (termed low eGFR) [1,2,3]. This CKD diagnostic stage often co-exists with disease comorbidities such as hypertension and proteinuria [2]. Albuminuria that persists for at least 3 months is also a CKD diagnostic criterion [1,2,3]. Albuminuria is designated, when the excretion of albumin (E_alb_), typically measured as the albumin-to-creatinine ratio (ACR), rises to levels above 20 and 30 mg/g creatinine in men and women, respectively [1,2,3].

Elevated risks of kidney damage [6,7,8], albuminuria [9,10,11], proteinuria [12,13], and CKD [14,15,16,17] have repeatedly been linked to chronic exposure to the metal pollutant cadmium (Cd) in many countries. There is also an increased mortality risk among CKD patients who had an elevated Cd body burden, reflected by Cd excretion rates ≥ 0.60 μg/g creatinine [18].

Cd is a metal contaminant with no nutritional value or physiological role, and it presents worldwide public health concerns because it is highly toxic [19]. For most people, exposure to Cd is unavoidable because it is found in most food types [20,21,22], cigarette smoke, and polluted air [23,24,25]. Most acquired Cd accumulates within the kidney tubular cells, where its levels increase through to the age of 50 years but decline thereafter, due to its release into the urine, as the injured tubular cells die [19,26]. Because most or all excreted Cd emanates from injured or dying tubular cells, excretion of Cd reflects the injury at the present time, not the risk of injury in the future [25,26].

Kidney tubular cell damage and tubular dysfunction, indicated by an increased excretion of the low-molecular weight protein, β_2_-microglobulin (β_2_M), are the most frequently reported adverse effects of Cd exposure [25]. An increase in β_2_M excretion above 300 µg/g creatinine was used in the toxicological risk assessment of dietary Cd [27,28]. However, current evidence has implicated the circulating β_2_M in blood pressure regulation [29], and β_2_M excretion above 300 µg/g creatinine is indicative of an increased risk of hypertension and severe kidney pathologies, such as rapid kidney functional deterioration and nephron loss for any reason [30,31,32].

The present study has three major objectives. The first is to examine the dose–response relationship between environmental Cd exposure levels and three adverse outcomes of such exposure, namely CKD, hypertension, and defective tubular function. Excretion of Cd (E_Cd_) and β_2_M (E_β2M_) were used as indicators of the long-term exposure or body burden of Cd and tubular dysfunction, respectively [25]. The second objective is to explore a connection between E_alb_ and the rising levels of systolic and diastolic blood pressure (SBP and DBP) in Cd-exposed people. The third objective is to address the female preponderance effects of environmental Cd exposure on blood pressure. These study objectives are formulated based on the current state of knowledge on the epidemiology of Cd toxicity and the significance of albuminuria as an independent risk factor for hypertension [33], a strong independent risk factor for worse outcomes of cardiovascular disease, incident CKD, and its progression to kidney failure, especially among those with type 2 diabetes [34,35,36,37].

## 2. Materials and Methods

### 2.1. Participant Selection

We assembled archived data from large Thai population-based cohorts of residents in the Mae Sot District, Tak Province, where environmental Cd contamination was endemic (*n* = 310), and two low-exposure areas in Bangkok (*n* = 192) and Nakhon-Si-Thammarat Province (*n* = 139).

The parent cohorts were conducted in compliance with the principles outlined in the Declaration of Helsinki [38,39]. The respective study protocols for the Mae Sot plus Bangkok groups and the Nakhon Si Thammarat group were approved by the Institutional Ethical Committees of Chulalongkorn University, Chiang Mai University, and the Mae Sot Hospital (Approval No. 142/2544, 5 October 2001) [38], as well as the Human Research Ethics Committee of Walailak University (Approval number WUEC-20-132-01, 28 May 2020) [39]. All participants gave informed consent prior to participation.

For all groups, the exclusion criteria were pregnancy, breast-feeding, a history of metal work, and a hospital record or physician’s diagnosis of an advanced chronic disease.

For the low-exposure groups, those aged 16 years or older were selected. The health status was ascertained by physician’s examination reports and routine blood and urinary chemistry profiles. For the Mae Sot group, those who had resided at their current addresses for 30 years or longer were selected. The sociodemographic data, educational attainment, occupation, health status, family history of diabetes, and smoking status were obtained by structured interview questionnaires.

The diagnosis of hypertension was based on the assessment made by the presiding physician and the recorded use of anti-hypertensive medication. Most of the participants with hypertension (94.9%) were being treated, while 5.1% were identified during a visit.

### 2.2. Assessment of Cadmium Exposure Levels and Its Effects

Samples of urine, whole blood, and plasma were collected from all participants after an overnight fast and were stored at −80 °C for later analysis. Plasma samples were assayed for the concentration of creatinine, while urine samples were assayed for the concentrations of creatinine, Cd, β_2_M, and alb, detailed previously [38,39].

The samples of soils and food crops from Nakhon Si Thammarat had arsenic, chromium, lead, and Cd within permissible ranges [40]. The risk of having diabetes type 2 among Nakhon Si Thammarat residents was not associated with the water arsenic concentration [41]. In comparison, the paddy soil samples from the Mae Sot district contained a Cd concentration exceeding the limit of 0.15 mg/kg, and household-storage rice samples contained four times the amount of the permissible Cd level of 0.1 mg/kg [42]. The urinary Cd excretion levels were correlated with the prevalence of hypertension and diabetes type 2 found among residents of the Mae Sot District (*n* = 5273) [43].

The eGFR was computed with equations of the Chronic Kidney Disease Epidemiology Collaboration (CKD-EPI) [44,45,46]. CKD stages 1, 2, 3, 4, and 5 corresponded to eGFRs of 90–119, 60–89, 30–59, 15–29, and <15 mL/min/1.73 m^2^, respectively [44,45].

### 2.3. Normalization of Cadmium, β_2_M, and Albumin Excretion Rates

The excretion of x (E_x_) was normalized to the creatinine clearance (C_cr_) as E_x_/C_cr_ = [x]_u_[cr]_p_/[cr]_u_, where x = Cd, β_2_M, or alb, [x]_u_ = urine concentration of x (mass/volume), [cr]_p_ = plasma creatinine concentration (mg/dL), and [cr]_u_ = urine creatinine concentration (mg/dL). E_x_/C_cr_ was expressed as an amount of x excreted per volume of the glomerular filtrate [47]. This C_cr_ normalization corrects for urine dilution and the number of functioning nephrons simultaneously, and it is not influenced by muscle mass.

The excretion of x (E_x_) was normalized to E_cr_ as [x]_u_/[cr]_u_, where x = Cd, β_2_M, or alb, [x]_u_ = urine concentration of x (mass/volume), and [cr]_u_ = urine creatinine concentration (mg/dL). E_x_/E_cr_ was expressed as an amount of x excreted per g of creatinine. This E_cr_ normalization corrects for urine dilution only. This method of normalization of the excretion rate is affected by interindividual differences in muscle mass, which produces non-differential errors, and a clear dose–response relationship of E_Cd_ and E_alb_ cannot be established [48,49].

### 2.4. Mediation Analysis for Cause–Effect Inference

A simple mediation model with one mediator (M) and the Sobel test for the statistical significance of an indirect effect of the independent variable X were as described by MacKinnon et al. (1995) and Preacher and Hayes (2004) [50,51,52].

A mediation model with M as a mediator of the effect of the independent variable X on the dependent variable Y and the standardized β coefficients describing the relationships of X, M, and Y are depicted below (Figure 1).

### 2.5. Statistical Analysis

The data were analyzed using IBM SPSS Statistics 21 (IBM Inc., New York, NY, USA). To assess the mean differences across E_β2M_ groups, the Kruskal–Wallis test was used. The Pearson chi-square test was used to assess the differences in percentages and prevalences of smoking, hypertension, low eGFR, diabetes, and albuminuria. The one-sample Kolmogorov–Smirnov test was used to ascertain the conformity to a normal distribution of continuous variables. Logarithmic transformation was applied to E_Cd_, E_β2M_, and E_alb_, which showed a right-skewed distribution. For eGFR, no data transformation was required, because the distribution of eGFR values was left-skewed. Multiple linear regression was conducted to define the predictors of E_alb_/C_cr_.

Logistic regression was conducted to evaluate the effects of Cd exposure and other independent variables on the prevalence odds ratio (POR) for CKD, hypertension, tubular dysfunction, and albuminuria. All reported POR values were adjusted for potential confounders.

Univariate analysis was used to obtain the covariate adjusted mean E_alb_/C_cr_ and eta square (η^2^), using the Bonferroni correction for comparing three eGFR groups. For all tests, *p*-values ≤ 0.05 were considered to indicate statistical significance.

## 3. Results

### 3.1. Cohort Participants

As the data in Table 1 indicate, the overall mean age of the cohort participants was 47.5 years, and the overall arithmetic (geometric) mean for the Cd excretion rate was 0.024 (0.009) µg/L filtrate, corresponding to 2.98 (1.11) µg/g creatinine. The % of smokers and diabetics were 29 and 10.8, respectively. Hypertension was the most prevalent (39.8%), followed by albuminuria (16.5%) and a low eGFR (4.8%).

To explore the potential effects of tubular dysfunction, participants were grouped by E_β2M_/C_cr_ values, and 422, 69, 61, and 61 participants were found to have E_β2M_/C_cr_ values of 1.0–2.9, 3.0–9.9, and ≥10 µg/L filtrate, respectively. The % hypertension, albuminuria, and low eGFR rose across E_β2M_/C_cr_ groups. In the highest E_β2M_/C_cr_ group, % hypertension, albuminuria, and a low eGFR were 62.3, 38.3, and 37.7, respectively. The corresponding % figures in the lowest E_β2M_/C_cr_ group were 34.4, 9.4, and 0.7. The mean SBP and mean DBP increased across E_β2M_/C_cr_ groups.

### 3.2. Effects of Cadmium on the Risks of Having CKD, Hypertension, and Tubular Malfunction

In three logistic regression analyses (Table 2), the prevalence odds ratio (POR) values for CKD, hypertension, and tubular dysfunction were not affected by gender or smoking, but the age, BMI, and diabetes did.

Age increased the risk of all three outcomes, while a rise in the BMI only increased the risk of hypertension. The risks of having CKD and hypertension rose 3.5-fold (*p* < 0.001) and 1.2-fold (*p* < 0.001), respectively, as the body burden of Cd increased two-fold. Doubling the Cd burden had little effect on the risk of having tubular dysfunction, defined as the E_β2M_/C_cr_ rates ≥ 3.0 µg/L filtrate [POR = 1.037 (95% CI: 0.917, 1.173), *p* = 0.567].

The tubular dysfunction was 17.7-time more prevalent among those with CKD. In comparison, all three outcomes were more prevalent among diabetics, compared with non-diabetics with the same overall Cd body burden.

### 3.3. Effects of Hypertension on the Prevalence of Albuminuria

In an inclusive logistic regression model (Table 3), the prevalence of albuminuria was minimally affected by the age, Cd body burden, being obese, gender, and smoking, while albuminuria was 3.3 times more prevalent in participants with CKD (*p* = 0.014).

Subgroup analysis revealed that hypertension was the key determinant of albuminuria in people with CKD, diabetes, and tubular dysfunction. The prevalence of albuminuria was found to be increased only in those with CKD and hypertension (POR = 4.3, *p* = 0.040). Similarly, respective prevalences of albuminuria were increased 5.4-fold, 2.9-fold, and 4.2-fold in those with diabetes plus hypertension, severe tubular dysfunction plus hypertension, and extremely severe tubular dysfunction plus hypertension.

### 3.4. Dose–Response Relationship and Quantitative Effect Size

As scatterplots indicate (Figure 1), a linear dose–response relationship of E_alb_/C_cr_ and E_Cd_/C_cr_ was found only in the high-Cd burden group, defined as E_Cd_/C_cr_ ≥ 0.01 µg/L filtrate, while a linear dose–response relationship of E_alb_/C_cr_ and blood pressure measures (SBP and DBP) existed in both the low- and high-Cd burden groups. E_alb_/C_cr_ was more closely associated with SBP than DBP in both Cd burden groups.

The results of the multiple linear regression analysis of E_alb_/C_cr_ (Table 4) indicate that age, BMI, eGFR, Cd burden, diabetes, blood pressure, gender, and smoking contributed to 14.7% and 11.3% of the total E_alb_/C_cr_ variation in the inclusive models 1 and 2, respectively.

In the SBP model including all subjects, the E_alb_/C_cr_ varied inversely with the eGFR (β = −0.180), while it varied directly with SBP (β = 0.263) and diabetes (β = 0.179). In the DBP model, the E_alb_/C_cr_ was inversely associated with the eGFR (β = −0.195), while it varied directly with DBP (β = 0.150) and diabetes (β = 0.220). In the DBP model only, the E_alb_/C_cr_ showed a positive association with the E_Cd_/C_cr_ (β = 0.122).

In the subgroup analysis, the E_alb_/C_cr_ was inversely associated with the eGFR only in women, β = −0.188 for the SBP model and β = −0.201 for the DBP model. An association of E_alb_/C_cr_ with DBP was also found in women only (β = 0.149). The E_alb_/C_cr_ was associated with SBP and diabetes in both women and men in the SBP model. Similarly, the E_alb_/C_cr_ was associated with diabetes in both women and men in the DBP model.

The influences of the Cd exposure levels and eGFR levels on albumin excretion among cohort participants are depicted in Figure 2, where the E_alb_/C_cr_ increased linearly with the SBP in all three eGFR subgroups (Figure 2a), but the increase in the E_alb_/C_cr_ with DBP was observed only in participants with CKD (Figure 2b).

In the univariate analysis for SBP effects with adjustment for covariates (Figure 2c), SBP contributed to 3.3% of the variation in E_alb_/C_cr_ in those with SBP higher than the median SBP value of 130 mmHg. The mean values for E_alb_/C_cr_ in participants with high SBP and CKD were 17.2% and 21.2% higher, compared to those with eGFR 61–90 and ≥90 mL/min/1.73 m^2^, respectively (Figure 2c).

In the univariate analysis for DBP effects with adjustment for covariates (Figure 2d), DBP contributed to 3.3% of the variation in E_alb_/C_cr_ in those with a DBP higher than the DBP median value of 80 mmHg. The mean values for E_alb_/C_cr_ in participants with high DBP and CKD were 19.0% and 22.5% higher, compared to those with eGFR 61–90 and ≥90 mL/min/1.73 m^2^, respectively (Figure 2d). Thus, the albumin excretion rate was increased in participants with CKD who also had a SBP and DBP higher than 130 and 80 mmHg, respectively.

A simple mediation analysis was conducted to assess the potential causal relationships of Cd exposure, eGFR and blood pressure levels (Figure 3).

A simple mediation analysis model of eGFR as the mediator suggested that Cd had direct and indirect effects on SBP and DBP (Figure 3a,b). However, the Sobel test results indicated that only the indirect effect of Cd on SBP reached a statistically significant level (Figure 3c,d). Thus, an increase in the SBP was in part due to reductions in the eGFR, and the proportion of Cd-induced SBP increment mediated by the eGFR decline was 80.6%. The rising DBP among participants was minimally related to Cd-induced eGFR reductions.

An additional simple mediation model with SBP or DBP as the mediator was conducted (Figure 4).

A direct effect of Cd on E_alb_/C_cr_ was not evident in the SBP model (β = 0.041, *p* = 0.381) or the DBP model (β = −0.015, *p* = 0.745) (Figure 4a,b). However, Cd had a significant indirect effect on the E_alb_/C_cr_ increment, mediated fully through rising SBP and DBP levels (Figure 4c,d). Thus, SBP and DBP were the full mediators of an elevation of the albumin excretion rate induced by Cd.

## 4. Discussion

The percentage (%) of hypertension among 641 cohort participants was the highest (39.8), followed by albuminuria (16.5), and low eGFR (4.8), while the percentages of smoking and diabetes were 29 and 10.8, respectively (Table 1). The % of hypertension and diabetes were in ranges with those reported for the representative U.S. population of 39 and 10.3–13, respectively [53,54]. The CKD (a low eGFR criterion) prevalence in this Thai cohort of 4.8% was lower than the 6.8% CKD prevalence figure in the Taiwanese population [55], but it was nearly half the prevalence of 9.3% found in a large U.S. population database, where 5175 CKD cases were identified from a total of 55,677 U.S. citizens, aged 20−85 years [15].

### 4.1. Low eGFR, Hypertension, and Albuminuria: Are They Causally Connected?

Albuminuria was 3.3-times more prevalent among participants with CKD, and the risks of having CKD and hypertension rose, respectively, 3.5-fold and 1.2-fold, when there was a two-fold increase in the body burden of Cd (Table 1 and Table 2). Hypertension was found to be the key determinant of albuminuria in participants with CKD; the risk of having albuminuria only increased in those with CKD plus hypertension (POR = 4.3) (Table 3).

We employed two simple mediation models to assess potential causal relationships of Cd exposure, eGFR, and blood pressure levels. In the model in which eGFR was the mediator (Figure 3), the increase in the SBP levels among the cohort participants was in part due to the reductions in eGFR. The proportion of the Cd-induced SBP increment mediated by the eGFR decline was 80.6%. The rising DBP among participants was minimally related to the Cd-induced eGFR reductions.

In the model in which SBP or DBP was the mediator (Figure 4), an elevation of the albumin excretion rate induced by Cd was through the rising SBP and DBP. Thus, albuminuria in Cd-exposed people appeared to be a consequence of a Cd-induced GFR fall, which causes the SBP and DBP to rise along with the excretion rate of albumin.

Herein, three outcomes of Cd exposure, namely a low eGFR, hypertension, and albuminuria, were investigated for dose–response and potential cause–effect inference. No previous studies have been undertaken to investigate the connection of these three outcomes. The cause-effect relationships of the Cd exposure levels, eGFR reductions, and blood pressure increment have been observed previously in a cohort of 447 Thai adults with a higher exposure level than participants in the present study [56]. The results suggested that Cd may increase blood pressure by destroying nephrons, which then caused the blood pressure to rise. This explains a universally high prevalence of hypertension in CKD patients.

A significant increase in the albumin excretion rate was found in participants with SBP and DBP within normal ranges (Figure 2). The mean values for the albumin excretion rate (E_alb_/C_cr_) among cohort participants who had a low eGFR and an SBP higher than the median 130 mmHg were 17.2% and 21.2% higher, compared to those with an eGFR 61–90 and ≥90 mL/min/1.73 m^2^, respectively (Figure 2). These results were obtained after adjustment for potential confounders. In addition, the lowest eGFR group with a DBP higher than the median of ≥ 80 mmHg excreted albumin at 19.0% and 22.5% higher rates than those with an eGFR 61–90 and ≥90 mL/min/1.73 m^2^, respectively. Intriguingly, in the SPRINT Trial (*n* = 2436), participants who had a low eGFR and an SBP of 16 mmHg higher than a mean value were found to have a 16% higher albumin excretion rate [57].

In the present study, the albumin excretion rate and albuminuria were analyzed as additional parameters including data from the Bangkok residents (a total 641 participants). The present study has first demonstrated a significant relationship between blood pressure and albumin excretion. This finding is of clinical significance because proteinuria (albuminuria) is a predictive of a continuing decline of the kidney function. Previously, we have analyzed albuminuria among Mae Sot residents with high-dose Cd exposure, resulting in overt tubular malfunction, where we found that Cd may impair the receptor mediated endocytosis for albumin [9]. It was estimated that Cd reduced the fractional reabsorption of albumin by 20%, if a glomerular sieving coefficient for albumin (GSC_alb_) of 10^−4^ was assumed [9].

### 4.2. Effects of Cadmium in Women and Men

By multiple regression analysis including all subjects, age, BMI, eGFR, Cd burden, diabetes, blood pressure, gender, and smoking contributed significantly to the variability of E_alb_/C_cr_ among cohort participants (Table 4). An inverse relationship between the E_alb_/C_cr_ and eGFR was found only in women (β = −0.188 for SBP model and β = −0.201 for DBP model). An association of E_alb_/C_cr_ with DBP was also found only in women (β = 0.149). The reasons for the female preponderance effect of Cd on blood pressure and albuminuria were not apparent from the present study. Further research is warranted.

Typically, hypertension is more prevalent in men, compared to age-matched premenopausal women [58,59,60], and the differences of the Cd effects in men and women were related to sex hormones. An increase in the urinary Cd excretion levels from <2 to ≥3 μg/g creatinine was associated with a 28% increase in serum testosterone in postmenopausal Japanese women [61]. An inverse association between urinary Cd and serum estradiol levels was noted in postmenopausal Japanese and Swedish women [62,63]. In the Swiss Kidney Project on Genes in Hypertension [64], the urinary Cd correlated with testosterone excretion in men, while there was a trend for an association in women.

### 4.3. Implications for the Toxicological Risk Assessment of Dietary Cadmium Exposure

Hypertension, albuminuria, and CKD have been found repeatedly in people with a low body burden of Cd. The risk of having hypertension doubled at a Cd excretion rate of 0.98 µg/g creatinine and a blood Cd level of 0.61 µg/L [56]. Similarly, an analysis of U.S. population data reported that the risk of having CKD increased 2.1-fold, 3.2-fold, and 5.5-fold in people who had blood Cd concentrations of 0.21–0.35, 0.36–0.60, and >0.60 µg/L, respectively [15].

Furthermore, given the same Cd body burden, the risk of having albuminuria was the highest (POR = 5.4) in diabetics with hypertension, compared to participants with severe tubular dysfunction plus hypertension (POR = 2.9) and participants with extremely severe tubular dysfunction plus hypertension (POR = 4.2) (Table 3). Thus, people with diabetes were highly susceptible to the nephrotoxicity of Cd. Consistent with our observation, a Dutch cross-sectional study, including 231 patients with type 2 diabetes, reported that low Cd exposure increased the risk of diabetic kidney disease [65,66]. In a six-year median follow-up of these patients, a progressive reduction of eGFR to kidney failure was linked to Cd exposure [61].

In the present study, albuminuria was 3.4-times more prevalent than a low eGFR. This underscores the utility of an elevated albumin excretion for early CKD detection purposes, given that CKD in its early stage is largely asymptomatic. This makes its early detection difficult and the initiation of early treatment, which can significantly prevent CKD progression, limited. An ACR as low as 7 mg/g creatinine was a predictor of incident CKD within 10 years [36]. An ACR ≥10 mg/g creatinine may increase mortality from all causes and CVD [37].

The strength of this study includes a high number of participants with diagnosed hypertension, as it meant that a reliable conclusion could be drawn from a cohort of a modest sample size (*n* < 1000). An additional strength includes the consideration given to a sample representative of the general population and the potential effects of smoking and diabetes in realistic population situations. A cross-sectional study design involving a one-time Cd exposure assessment was a limitation. Because of the difference in hormonal status, including menopausal and post-menopausal women in this study was another limitation.

## 5. Conclusions

Through the mediation analysis, this study shows, for the first time, that an elevation in the albumin excretion rate is caused by an increase in blood pressure, notably SBP, which arises from Cd-induced GFR reductions. A declining eGFR and rising SBP can serve as sensitive endpoints suitable for the health risk assessment of exposure to dietary Cd and the derivation of the health-protective exposure limit.

## Data Availability

All data are contained within this article.

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
