# Peer review of "Albuminuria in People Chronically Exposed to Low-Dose Cadmium Is Linked to Rising Blood Pressure Levels"

_toxics, 2025, doi:10.3390/toxics13020081_

Round 1

Reviewer 1 Report

Comments and Suggestions for Authors

This study examines the mediating effects of cadmium exposure on the development of albuminuria, via blood pressure and chronic kidney disease, as health effects of cadmium. However, there are few significant results on the association between body burden of cadmium and albuminuria or urinary albumin concentration. I consider that several modifications are necessary for acceptance of publication.

1.    Lines 379-380, ‘The extent of albumin excretion could thus serve as a sensitive toxicity endpoint for the toxicological risk assessment of Cd in the human diet.’ and lines 387-388 ‘and an elevated albumin excretion‘ : The results of this study do not consistently show a strong association between urinary cadmium concentrations and the presence of albuminuria or elevated urinary albumin concentrations. Therefore, the results of this study do not clearly support these statements. These should be deleted.

2.    Lines 380-383, ‘Current dietary Cd exposure guidelines, based on a severe tubular dysfunction (Eβ2M/Ccr ≥ 3 μg/L filtrate or β2M excretion rates ≥ 300 μg/g creatinine), are not protective of human health. There is an urgent need to develop new dietary Cd exposure limits.’: The insignificant results of this study do not support that the criteria for renal tubular damage (Eβ2M/Ccr ≥ 3 μg/L filtrate or β2M excretion rate ≥ 300 μg/g creatinine) are not protective enough or not valid for human health. These statements should be deleted.

Author Response

Reviewer 1

Comments and Suggestions

This study examines the mediating effects of cadmium exposure on the development of albuminuria, via blood pressure and chronic kidney disease, as health effects of cadmium. However, there are few significant results on the association between body burden of cadmium and albuminuria or urinary albumin concentration. I consider that several modifications are necessary for acceptance of publication.

  1. Lines 379-380, ‘The extent of albumin excretion could thus serve as a sensitive toxicity endpoint for the toxicological risk assessment of Cd in the human diet.’ and lines 387-388 ‘and an elevated albumin excretion‘: The results of this study do not consistently show a strong association between urinary cadmium concentrations and the presence of albuminuria or elevated urinary albumin concentrations. Therefore, the results of this study do not clearly support these statements. These should be deleted.

RESPONSE: We thank the reviewer for evaluating our work and for raising this issue of concern.  Accordingly, the statement regarding albumin excretion has been deleted.  The remainder statement in the conclusion now reads below (lines 405-407).

“A declining eGFR and rising SBP can serve as sensitive endpoints suitable for toxicological risk assessment of Cd in the human diet and the derivation of health-protective exposure limit.” 

  1. Lines 380-383, ‘Current dietary Cd exposure guidelines, based on a severe tubular dysfunction (Eβ2M/Ccr ≥ 3 μg/L filtrate or β2M excretion rates ≥ 300 μg/g creatinine), are not protective of human health. There is an urgent need to develop new dietary Cd exposure limits.’: The insignificant results of this study do not support that the criteria for renal tubular damage (Eβ2M/Ccr ≥ 3 μg/L filtrate or β2M excretion rate ≥ 300 μg/g creatinine) are not protective enough or not valid for human health. These statements should be deleted.

RESPONSE:  We have eliminated all sentences the reviewer has referred to.  To better reflect the novelty of our study, two new paragraphs have been inserted in the Discussion (lines 318-333), quoted below.

“Herein, three outcomes of Cd exposure, namely a low eGFR, hypertension and albuminuria were investigated for dose-response and potential cause-effect inference. No previous studies have been undertaken to investigate the connection of these three outcomes. The relationships of Cd exposure levels, eGFR reductions and blood pressure increment have been observed previously in a cohort of 447 Thai adults with a higher exposure level than participants in the present study [56]. The results suggested that an increase in blood pressure may be a consequence of a decrease in GFR induced by Cd. This observation explains why patients with CKD have hypertension almost universally.

In the present study, excretion of albumin and albuminuria were analyzed as additional parameters including also data from the Bangkok residents (a total 641 participants). To the best of our knowledge, the present study has provided, for the first time, a direct relationship between blood pressure and albumin excretion.  This finding is of clinical significance because proteinuria (albuminuria) is a strong predictor of continued progressive functional decline of the kidney. Intriguingly, in the SPRINT Trial (n = 2436), participants who had a low eGFR and SBP of 16 mmHg higher than a mean value were found to have a 16% higher albumin excretion rate [57].”

[56] Satarug, S.; Vesey, D.A.; Yimthiang, S.; Khamphaya, T.; Pouyfung, P.; Đorđević, A.B. Environmental Cadmium Exposure Induces an Increase in Systolic Blood Pressure by Its Effect on GFR. Stresses 2024, 4, 436-451.

[57] Ikeme, J.C.; Katz, R.; Muiru, A.N.; Estrella, M.M.; Scherzer, R.; Garimella, P.S.; Hallan, S.I.; Peralta, C.A.; Ix, J.H.; Shlipak, M.G. Clinical Risk Factors For Kidney Tubule Biomarker Abnormalities Among Hypertensive Adults With Reduced eGFR in the SPRINT Trial. Am. J. Hypertens. 2022, 35, 1006-1013.

Reviewer 2 Report

Comments and Suggestions for Authors

There are a few issues that the authors need to clarify:

1. Give more information about the endemic Cd contamination - Cd content in soil, water, etc. Were the people from that region exposed to a low-dose but chronic Cd exposure?

2. Why are not the results presented by regions? Are there no differences in the parameters examined between the cohorts from the 3 different regions?

3. What type is the diabetes?

4. Was Cd content in urine similar in the cohorts from the 3 regions included in the study?

5. If the risk of developing CKD rises significantly as body burden of Cd increases how do the authors explain the little effect of Cd on tubular dysfunction as tubular dysfunction is prevelant among those with CKD?

6. In the paper of Satarug et al. 2022, authored by some of the co-authors of the present manuscript, in a similar study they find that "normalization of Cd excretion to Ccr unravels the adverse effect of Cd on GFR and provides a more accurate evaluation of the severity of the tubulo-glomerular effect of Cd". Does this mean that Eb2M/Ccr is not a reliable marker for tubular dysfunction in case of Cd intoxication?

Author Response

Reviewer 2.

Comments and Suggestions

There are a few issues that the authors need to clarify:

Point 1. Give more information about the endemic Cd contamination - Cd content in soil, water, etc. Were the people from that region exposed to a low-dose but chronic Cd exposure?

RESPONSE:  We have now provided data on Cd concentrations found in environmental samples (lines 115-122), quoted below with 4 more references.  Changes to the text in a manuscript are in blue.

“Levels of arsenic, chromium, lead, and Cd in the samples of soils and food crops in Nakhon Si Thammarat were within permissible ranges [40]. No association was found between water arsenic concentration and the risk of diabetes [41]. In comparison, the Cd concentration of the paddy soil samples from the Mae Sot district exceeded the limit of 0.15 mg/kg, and the rice samples collected from household storage contained four times the amount of the permissible Cd level of 0.1 mg/kg [42]. In a health survey of residents of the Mae Sot District (n = 5273), urinary Cd excretion levels correlated with the prevalence of hypertension and diabetes type 2 [43].”

[40] Zarcinas, B.A.; Pongsakul, P.; McLaughlin, M.J.; Cozens, G. Heavy metals in soils and crops in Southeast Asia. 2. Thailand. Environ. Geochem. Health 2004, 26, 359–371.

[41] Sripaoraya, K.; Siriwong, W.; Pavittranon, S.; Chapman, R.S. Environmental arsenic exposure and risk of diabetes type 2 in Ron Phibun subdistrict, Nakhon Si Thammarat Province, Thailand: Unmatched and matched case-control studies. Risk Manag Healthc. Policy 2017, 10, 41–48.

[42] Suwatvitayakorn, P.; Ko, M.S.; Kim, K.W.; Chanpiwat, P. Human health risk assessment of cadmium exposure through rice consumption in cadmium-contaminated areas of the Mae Tao sub-district, Tak, Thailand. Environ. Geochem. Health 2020, 42, 2331–2344.

[43] Swaddiwudhipong, W.; Mahasakpan, P.; Limpatanachote, P.; Krintratun, S. Correlations of urinary cadmium with hypertension and diabetes in persons living in cadmium-contaminated villages in northwestern Thailand: A population study. Environ. Res. 2010, 110, 612–616.

Point 2. Why are not the results presented by regions? Are there no differences in the parameters examined between the cohorts from the 3 different regions?

RESPONSE: The main purpose of the present study was to examine the mediating effects of Cd exposure on the development of albuminuria, via rising blood pressure, and CKD (a low eGFR), as health effects of Cd.  We quantified tubular cell damage, tubular proteinuria, SBP and DBP according to the eGFR and Cd exposure levels in those diagnosed with and without hypertension.

To obtain a group of people suitable for dose-response and mediation analyses, data were collected from cohorts of residents of three locations with differences in prevalence of hypertension, and diabetes type 2. Because the residents recruited to the present study resided in the zones with modest Cd contamination of the Mae Sot District, their chronic Cd exposure levels were much lower than those resided in the hot spots described in ref. 41.  Based on the overall geometric mean urinary Cd of 1.11 µg/g creatinine, and age ranging between 16 and 80 years (mean 47.5), we believe that this group of 641 persons could be considered as the representative of the general Thai population with chronic low-to-moderate exposure to environmental Cd. 

Point 3. What type is the diabetes?

RESPONSE: We have now specified type of diabetes in the text.

Point 4. Was Cd content in urine similar in the cohorts from the 3 regions included in the study?

RESPONSE: Please see response to Point 2 above.  

Point 5. If the risk of developing CKD rises significantly as body burden of Cd increases how do the authors explain the little effect of Cd on tubular dysfunction as tubular dysfunction is prevalent among those with CKD?

Point 6. In the paper of Satarug et al. 2022, authored by some of the co-authors of the present manuscript, in a similar study they find that "normalization of Cd excretion to Ccr unravels the adverse effect of Cd on GFR and provides a more accurate evaluation of the severity of the tubulo-glomerular effect of Cd". Does this mean that Eβ2M/Ccr is not a reliable marker for tubular dysfunction in case of Cd intoxication?

RESPONSE: Thank you for these insightful comments and guidance.  The reviewer has correctly stated that Eβ2M/Ccr is not a reliable marker for tubular dysfunction in case of Cd intoxication. We have shown that when a low eGFR was used as a CKD diagnostic criterion and when ECd was normalized to Ccr (Satarug et al. 2022), a fall of eGFR was found to be an early effect of Cd exposure. Accordingly, two new paragraphs, quoted below, have been inserted (lines 318-333) in the Discussion to reflect the novelty of the present study.

“Herein, three outcomes of Cd exposure, namely a low eGFR, hypertension and albuminuria were investigated for dose-response and potential cause-effect inference. No previous studies have been undertaken to investigate the connection of these three outcomes. The relationships of Cd exposure levels, eGFR reductions and blood pressure increment have been observed previously in a cohort of 447 Thai adults with a higher exposure level than participants in the present study [56]. The results suggested that an increase in blood pressure may be a consequence of a decrease in GFR induced by Cd. This observation explains why patients with CKD have hypertension almost universally.

In the present study, excretion of albumin and albuminuria were analyzed as additional parameters including also data from the Bangkok residents (a total 641 participants). To the best of our knowledge, the present study has provided, for the first time, a direct relationship between blood pressure and albumin excretion.  This finding is of clinical significance because proteinuria (albuminuria) is a strong predictor of continued progressive functional decline of the kidney. Intriguingly, in the SPRINT Trial (n = 2436), participants who had a low eGFR and SBP of 16 mmHg higher than a mean value were found to have a 16% higher albumin excretion rate [57].”

[56] Satarug, S.; Vesey, D.A.; Yimthiang, S.; Khamphaya, T.; Pouyfung, P.; Đorđević, A.B. Environmental Cadmium Exposure Induces an Increase in Systolic Blood Pressure by Its Effect on GFR. Stresses 2024, 4, 436-451.

[57] Ikeme, J.C.; Katz, R.; Muiru, A.N.; Estrella, M.M.; Scherzer, R.; Garimella, P.S.; Hallan, S.I.; Peralta, C.A.; Ix, J.H.; Shlipak, M.G. Clinical Risk Factors For Kidney Tubule Biomarker Abnormalities Among Hypertensive Adults With Reduced eGFR in the SPRINT Trial. Am. J. Hypertens. 2022, 35, 1006-1013.

Reviewer 3 Report

Comments and Suggestions for Authors

My main observation regarding the manuscript refers to a publication I found (belonging to the same authors) - Satarug, S.; Vesey, D.A.; Yimthiang, S.; Khamphaya, T.; Pouyfung, P.; Đorđević, A.B. Environmental Cadmium Exposure Induces an Increase in Systolic Blood Pressure by Its Effect on GFR. Stresses 20244, 436-451. https://doi.org/10.3390/stresses4030029.

In this paper, the authors  "quantified tubular cell damage, tubular proteinuria, systolic, and diastolic blood pressures (SBP and DBP) according to the estimated GFR (eGFR) and Cd exposure levels in those diagnosed with and without hypertension".

As the two subjects are related and quite similar, the authors should include a section that mentions the results of their previous publication and emphasize the novelty of the new study.

Other comments are:

  • Check the formula at line 119 - [Cd]u[cr]p/[cr]u or [X]u[cr]p/[cr]u?
  • Check diagram 1 - C' indirect/direct effect of X on Y.

Author Response

Reviewer 3

Comments and Suggestions

My main observation regarding the manuscript refers to a publication I found (belonging to the same authors) - Satarug, S.; Vesey, D.A.; Yimthiang, S.; Khamphaya, T.; Pouyfung, P.; Đorđević, A.B. Environmental Cadmium Exposure Induces an Increase in Systolic Blood Pressure by Its Effect on GFR. Stresses 2024, 4, 436-451. https://doi.org/10.3390/stresses4030029.

In this paper, the authors "quantified tubular cell damage, tubular proteinuria, systolic, and diastolic blood pressures (SBP and DBP) according to the estimated GFR (eGFR) and Cd exposure levels in those diagnosed with and without hypertension".

As the two subjects are related and quite similar, the authors should include a section that mentions the results of their previous publication and emphasize the novelty of the new study.

RESPONSE: We thank the reviewer for comments and suggestions. The issues the reviewer raised has been addressed in a new paragraph (lines 318-333) quoted below.  Changes to the text are in blue.

“Herein, three outcomes of Cd exposure, namely a low eGFR, hypertension and albuminuria were investigated for dose-response and potential cause-effect inference. No previous studies have been undertaken to investigate the connection of these three outcomes. The relationships of Cd exposure levels, eGFR reductions and blood pressure increment have been observed previously in a cohort of 447 Thai adults with a higher exposure level than participants in the present study [56]. The results suggested that an increase in blood pressure may be a consequence of a decrease in GFR induced by Cd. This observation explains why patients with CKD have hypertension almost universally.

In the present study, excretion of albumin and albuminuria were analyzed as additional parameters including also data from the Bangkok residents (a total 641 participants). To the best of our knowledge, the present study has provided, for the first time, a direct relationship between blood pressure and albumin excretion.  This finding is of clinical significance because proteinuria (albuminuria) is a strong predictor of continued progressive functional decline of the kidney. Intriguingly, in the SPRINT Trial (n = 2436), participants who had a low eGFR and SBP of 16 mmHg higher than a mean value were found to have a 16% higher albumin excretion rate [57].”

[56] Satarug, S.; Vesey, D.A.; Yimthiang, S.; Khamphaya, T.; Pouyfung, P.; Đorđević, A.B. Environmental Cadmium Exposure Induces an Increase in Systolic Blood Pressure by Its Effect on GFR. Stresses 2024, 4, 436-451.

[57] Ikeme, J.C.; Katz, R.; Muiru, A.N.; Estrella, M.M.; Scherzer, R.; Garimella, P.S.; Hallan, S.I.; Peralta, C.A.; Ix, J.H.; Shlipak, M.G. Clinical Risk Factors For Kidney Tubule Biomarker Abnormalities Among Hypertensive Adults With Reduced eGFR in the SPRINT Trial. Am. J. Hypertens. 2022, 35, 1006-1013.

Other comments are:

Check the formula at line 119 - [Cd]u[cr]p/[cr]u or [X]u[cr]p/[cr]u?

RESPONSE: A correction has been undertaken. An equation now reads, [X]u[cr]p/[cr]u.

Check diagram 1 - C' indirect/direct effect of X on Y.

RESPONSE:  An error has been corrected. The C’ has been changed to “Direct effect of X on Y”.

Reviewer 4 Report

Comments and Suggestions for Authors

This manuscript uses statistical analysis of a Thai database to analysis associations between cadmium and several health conditions.  The manuscript is well written and reads easily.  The experimental details necessary are provided, and the results are presented clearly and properly.  The association between cadmium and the title conditions are important so that the manuscript would make a meaningful contribution to the literature.  Because of the above, I have only a few minor comments:  

First, the title should be changed from "Due to" to "linked to" or something similar as this type of statistical study only discloses potential links and does not establish cause.

Avoid the use of first person voice (e.g., "we") except in the introduction and conclusion.

Try to avoid one sentence paragraphs in the results section.

A paragraph at the end of the manuscript discussing the limitations of the study would be helpful to readers.

Author Response

Reviewer 4

Comments and Suggestions for Authors

This manuscript uses statistical analysis of a Thai database to analysis associations between cadmium and several health conditions.  The manuscript is well written and reads easily.  The experimental details necessary are provided, and the results are presented clearly and properly.  The association between cadmium and the title conditions are important so that the manuscript would make a meaningful contribution to the literature.  Because of the above, I have only a few minor comments:  

Comment 1. First, the title should be changed from "Due to" to "linked to" or something similar as this type of statistical study only discloses potential links and does not establish cause.

RESPONSE:  The title has been changed to, “Albuminuria in People Chronically Exposed to Low-Dose Cadmium is Linked to Rising Blood Pressure Levels.”

Comment 2. Avoid the use of first person voice (e.g., "we") except in the introduction and conclusion. Try to avoid one sentence paragraphs in the results section.

RESPONSE:  Use of an active voice has been limited to the introduction and conclusion.  All paragraphs in the results section have now been reconstructed to contain multiple sentences.

Comment 3 A paragraph at the end of the manuscript discussing the limitations of the study would be helpful to readers.

RESPONSE:  A new paragraph has been inserted to declare strength and limitation of the present study (lines 395-401), quoted below.

“The strength of this study includes a high number of participants with diagnosed hypertension as it meant that a reliable conclusion could be drawn from a cohort of modest sample size (n < 1000). An additional strength includes the consideration given to potential effects of smoking and diabetes in realistic population situations. A one-time-only assessment of Cd exposure and its outcomes was a limitation. Including both menopausal and post-menopausal women was another limitation because of their hormonal status differences.”

Round 2

Reviewer 1 Report

Comments and Suggestions for Authors

The manuscript has been revised enough for publication.

Reviewer 3 Report

Comments and Suggestions for Authors

I have no further comments.